# Frailty as an Independent Predictor of Adverse Outcomes in Patients Undergoing Direct Myocardial Revascularization

**DOI:** 10.3390/diagnostics14131419

**Published:** 2024-07-03

**Authors:** Kristina Krivoshapova, Daria Tsygankova, Evgeny Bazdyrev, Olga Barbarash

**Affiliations:** Federal State Budgetary Institution “Research Institute for Complex Issues of Cardiovascular Diseases”, Kemerovo 650002, Russia; darjapavlovna2014@mail.ru (D.T.); edb624@mail.ru (E.B.); reception@kemcardio.ru (O.B.)

**Keywords:** frailty, frail patients, coronary artery disease, coronary artery bypass grafting, adverse outcome

## Abstract

The aim of this study was to analyze the incidence and anamnestic characteristics of frail patients with stable coronary artery disease (CAD) and to evaluate the role of frailty in the development of complications and adverse outcomes in the perioperative period and early survival period after coronary artery bypass grafting (CABG). Material and methods: The study included 387 patients admitted to the clinic for a scheduled primary CABG. A seven-item questionnaire, “PRISMA-7”, was used to identify frail elderly patients before the procedure. We divided the study sample into two groups, taking into account the results of the survey: patients without frailty, n_0_ = 300 (77.5%), and patients with frailty, n_1_ = 87 (22.5%). The anamnestic and laboratory data, outcome of the surgical intervention, perioperative and early complications, and adverse outcomes were analyzed. Results: We detected frailty in 22.5% of the patients with CAD before the procedure. According to the anamnestic data and paraclinical and intraoperative findings, the groups of patients with and without frailty were comparable. The differences were revealed in the intraoperative and early postoperative periods of CABG. Thus, postoperative rhythm disturbances (19.5% vs. 10.5%, *p* = 0.025, V = 0.115, respectively) and transient ischemic attacks/stroke (5.7% vs. 1.3%, *p* = 0.031, V = 0.122, respectively) occurred significantly more often among the frail patients. There were no significant differences between the groups in the frequency of other intraoperative and early postoperative complications. In the group of frail patients, four fatal outcomes due to early postoperative ischemia were recorded, and among patients without frailty, one fatal outcome was recorded (4.5% vs. 0.3%, *p* = 0.010, V = 0.156, respectively). At the 1-year follow-up visit, the presence of frailty in history served as a predictor of mortality (11.5% vs. 0.6%, *p* ˂ 0.001, V = 0.290, respectively). Conclusion: The presence of frailty can be used as an independent predictor of an unfavorable prognosis in patients with CAD, both in the perioperative and early survival period after CABG. It should be taken into account during surgical risk assessment.

## 1. Introduction

Coronary artery bypass grafting (CABG) is one of the most effective treatment modalities for coronary artery disease (CAD) [1]. Over the past few years, the number of elderly and senile patients being operated on has increased, and their “clinical picture” has changed: the number of patients with severe CAD at high risk of developing cardiovascular complications due to comorbidity has increased [2].

Specialists in the field of cardiac surgery have developed various prognostic systems, models, and scales to assess the risk of postoperative complications and adverse outcomes of surgery [3]. The world’s best cardiac surgery centers have implemented many of them successfully. However, due to false positive and false negative results, one should take into account the limitations associated with their use [4]. The most recognized and widespread models of assessing cardiac risk are the European System for Cardiac Operative Risk Evaluation I/II (EuroSCORE I/II) [5,6], the Society of Thoracic Surgeons 2008 Cardiac Surgery Risk Model (STS) score [7], and the Cleveland Clinic scoring system [8]. At the same time, it should be noted that the scoring systems have been developed within different time frames and on different clinical and demographic indicators of patients, and use different criteria for assessing cardiovascular complications. Thus, developing a tool to effectively assess postoperative prognosis in CAD patients before myocardial revascularization seems highly relevant.

Over the past few years, interest in studying the characteristics of “frail” patients in the perioperative period of CABG has been growing exponentially. Frailty syndrome (FS) in the elderly is characterized by its multidimensionality, involving the decline of physical, biological, social, and psychological domains, which damage homeostatic reserves and, therefore, increase vulnerability to stressors [9]. According to a number of studies, FS in the preoperative period is directly correlated with a high risk of complications and adverse outcomes. In epidemiological studies, prospective registries, observational studies, and systematic reviews, FS is associated with a twofold increase in the risk of severe complications, repeated hospitalizations, and mortality [10,11]. Moreover, according to the research data, the duration of hospitalization and the cost of care and rehabilitation are significantly higher in frail elderly patients (by 15–60% on average) [12,13]. In the preoperative period, it is necessary to detect FS since it can serve as a reflection of the state of a person’s health. However, specialists still have not agreed upon a single pathogenesis of FS, methods for routine diagnosis, recommendations for perioperative rehabilitation, or methods for the prevention of complications and adverse outcomes in frail patients undergoing surgery.

All of the above-mentioned factors serve as a prerequisite for conducting the current study aimed at assessing the incidence of FS, identifying the clinical characteristics of frail patients with stable coronary artery disease before surgery, and the impact of FS on the perioperative period and early survival period after CABG.

## 2. Materials and Methods

The registry-based observational study was conducted from January 2018 to June 2019 at the Cardiology Department of the Research Institute for Complex Issues of Cardiovascular Diseases (Kemerovo); the study included 1400 patients with coronary artery disease who were admitted to the clinic for a planned primary CABG. Taking into account the inclusion and exclusion criteria, the final analysis included 387 patients who provided informed consent to participate in this study. The diagnosis of coronary artery disease was verified using the ESC-2019 Guidelines on Chronic Coronary Syndromes (stable coronary artery disease). The functional class (FC) of angina pectoris was assessed using the classification of the Canadian Cardiovascular Society (CCS, 1976). The classification created by V.H. Vasilenko and N.D. Strazhesco (1935) was used to assess heart failure. The FC of HF was evaluated according to the New York Heart Association classification (NYHA, 1964). The primary screening of frail patients before surgery was performed using one of the many tools used for the diagnosis of FS worldwide—the questionnaire “PRISMA-7” (Figure 1) [14].

The inclusion criteria were as follows: the patient’s consent to participate in the study; planned primary CABG; and absence of exclusion criteria. The exclusion criteria were as follows: acute coronary syndrome; simultaneous interventions on heart valves and great vessels of the heart; decompensated heart failure (HF); uncontrolled arterial hypertension (AH); inflammatory diseases; severe chronic obstructive pulmonary disease; type I diabetes mellitus (DM); stage IV-V chronic kidney disease (or GFR of less than 30 mL/min); alcoholism; central nervous system diseases; traumatic brain injuries; taking a number of medications (oral steroids, antidepressants, barbiturates, and muscle relaxants); inability to understand and (or) perform the tasks mentioned in the study protocol; and refusal to participate or continue participation in the study.

Our study took into account asymptomatic or accompanied-by-complaint episodes of cardiac arrhythmia that occurred for the first time after surgery. Ventricular arrhythmias were assessed according to the Lown classification and were represented by high-grade ventricular extrasystole (III, IV-A, IV-B). Ventricular rhythm disturbances recorded during the day and at night were in most cases asymptomatic and did not manifest themselves clinically. All patients included in the study maintained stable hemodynamics during ventricular arrhythmias. All paroxysmal AFib were tachysystolic with an average ventricular rate from 110 to 160 beats/min, accompanied by shortness of breath, dizziness, chest discomfort, a drop in blood pressure, and weakness. There were no cases of syncope. Acute cerebrovascular accident was defined as focal brain damage, detected clinically and confirmed by computed tomography.

From 2019 to 2020, a one-year follow-up was carried out by telephone and/or by analyzing the data on hospitalizations, new diagnoses, and invasive medical procedures (all-cause mortality, myocardial infarction, ischemic stroke, repeated hospitalizations for decompensated HF and unstable angina, and revascularization by percutaneous coronary intervention).

Statistical processing of the results was carried out using IBM SPSS Statistics 26.0.0 software. Absolute and relative values (%) were used to describe qualitative features. The normality of the distribution of quantitative features was assessed using the Shapiro–Wilk criterion. Quantitative features are represented by the median and interquartile range (Me [Q1; Q3]). Pearson’s criterion χ^2^ and Fisher’s exact test were used to assess the statistical significance of the differences in the qualitative characteristics of the two independent groups, and the strength of the relationship was estimated using Cramer’s V. The Mann–Whitney criterion was used to compare the two independent groups. A predictive model was developed using binary logistic regression (Wald’s method). To evaluate the performance of a predictive model we used a ROC curve, which was defined as a plot of test sensitivity as the y coordinate versus its 1-specificity or false positive rate as the x coordinate. The quality of a predictive model was assessed based on the area under the ROC curve with standard error and 95% confidence interval (CI). The differences were considered statistically significant at *p* ≤ 0.050.

## 3. Results

The analysis of the baseline clinical and anamnestic characteristics of the sample reveals a classic clinical picture of patients with coronary artery disease undergoing CABG. The mean age was 65.0 [59.0; 69.0] years, and the majority were men (73.1%). Most of the patients had FC I-II angina pectoris (78.8%), NYHA FC I-II HF (91.7%), and hypertension in their history (83.5%). Postinfarction cardiac sclerosis (PICS) was detected in 57.1% of the patients. Peripheral artery disease (PAD) (32.0%) and atherosclerosis of the extracranial brachiocephalic arteries (58.1%) were the most common concomitant pathologies detected in the patients’ history, and one-quarter of the patients had type II diabetes in their history (Table 1). In most cases of atherosclerosis of the brachiocephalic arteries, the damage to the brachiocephalic vessels was hemodynamically insignificant or there was “borderline” stenosis, requiring monitoring using ultrasound and continuous follow-up with a neuroradiologist specialist. In 27.0% of patients, carotid endarterectomy (CEE) was planned as a second stage of treatment.

We divided the study sample into two groups, taking into account the results of the survey: patients without FS, n_0_ = 300 (77.5%), and patients with FS, n_1_ = 87 (22.5%). The groups were comparable in age (Table 2). FS was significantly more common in men (*p* = 0.008). The results of the statistical analysis revealed a tendency of patients with FS to present with a higher severity of cardiovascular comorbidities. Thus, patients with FS were more likely to present with a history of PICS (67.0% and 54.2%, *p* = 0.032, respectively). Despite the fact that cerebral atherosclerosis was more common in patients without FS (62.9% and 50.0%, *p* = 0.030, respectively), PAD was detected more frequently in frail patients (42.0%) compared to patients without FS (29.1%, *p* = 0.022) (Table 2).

The groups were comparable in most paraclinical characteristics in the preoperative period of CABG, with the exception of a higher level of low-density lipoproteins (2.7 [2.1; 3.6] and 2.5 [1.8; 3.2], *p* = 0.026, respectively) and total cholesterol (4.6 [3.8; 5.8] and 4.5 [3.5; 5.3], *p* = 0.008, respectively) in the group of patients without FS.

CABG with cardiopulmonary bypass (CPB) was performed according to the standard protocols adopted at the Research Institute. The duration of the surgery and CPB, mechanical ventilation (MV), and length of stay in the ICU in the comparison groups did not differ significantly. The groups did not differ in the duration of aortic cross-clamping, the number of grafts, or the volume of blood loss on the first day after surgery. More details about the surgery are presented in Table 3.

The analysis of the incidence of adverse outcomes and complications in the intraoperative and early postoperative periods after surgery in the groups revealed significant differences. As expected, the patients with FS presented significantly more often with rhythm disturbances (19.5% and 10.5%, *p* = 0.025, V = 0.115, respectively) and TIA/stroke (5.7% and 1.3%, *p* = 0.031, V = 0.122, respectively) compared to the patients without FS. Moreover, four deaths were recorded in the frail patients due to cardiac ischemic events in the early postoperative period of CABG, whereas only one death occurred in the group without FS (4.5% and 0.3%, *p* = 0.010, V = 0.156, respectively). There were no significant differences between the groups in the frequency of other complications in the early postoperative period of CABG (Table 4).

The risk of mortality in the early postoperative period of CABG in the FS patients increased by 5.25 times (95% CI, 2.16–19.13) due to cardiac ischemic events. FS also increased the risk of TIA/stroke by 4.48 times (95% CI, 1.18–17.07), and rhythm disturbances by 2.08 times (95% CI, 1.09–3.97). Generally, the frail patients were two times more likely (95% CI, 1.34–3.88) to develop complications in the early postoperative period of CABG.

The 1-year follow-up revealed no differences in the number of endpoints. Despite this, the number of deaths due to cardiovascular pathology that developed within a year after CABG was significantly higher in the frail patients: two deaths (0.6%) occurred in the group of patients without FS, and ten deaths (11.5%) occurred in the group of patients with FS, *p* < 0.001, OR 6.92, and CI 95%, 2.25–15.25 (Table 4).

Having conducted an analysis of the risk, we built a model to predict mortality in FS patients. To predict the dependent variable based on the independent variables, we used Nagelkerke R-squared. ROC curve analysis was used to assess the prognostic significance. The cut-off point was determined using the highest value of the Youden index.

Using binary logistic regression, the prognostic model should determine the risk of fatal outcome within the first year after CABG, depending on the presence of FS according to the “PRISMA-7” questionnaire. This dependence can be described by the following equation:p = 1/(1 + e^−z^) × 100%
z = −5.690 + 3.719X_FS_,
where p is the risk of mortality within the first year after CABG, and X_FS_ is FS according to the “PRISMA-7” questionnaire (0—no FS; 1—FS).

The obtained regression model was statistically significant (*p* < 0.001). Based on the Nagelkerke modification, this model explained 27.7% of the observed variance in the risk of a fatal outcome within the first year after CABG. When assessing the variable “FS according to the “PRISMA-7” questionnaire”, the risk of mortality within a year after CABG increases by 4.21 times in the presence of FS (95% CI, 2.24–17.08).

We obtained the following curve when assessing the dependence of the risk of mortality within the first year after the CABG from the value of the logistic function p using ROC curve analysis (Figure 2).

The area under the ROC curve was 0.853 ± 0.070 with 95% CI: 0.716–0.990. The model was statistically significant (*p* < 0.001). The threshold value of the logistic function p at the cut-off point, which corresponds to the highest value of the Youden index, was 0.122. The risk of a fatal outcome within the first year after the CABG was predicted to be greater than or equal to the value of the logistic function p. The sensitivity and specificity of the model were 73.7% and 72.9%, respectively.

## 4. Discussion

According to the results of a survey of CAD patients before CABG, FS was detected in 22.5% of the patients. The presence of FS in this study was associated with comorbidities like PICS (*p* = 0.032) and PAD (*p* = 0.022), thus confirming higher incidence rates of comorbidities in frail patients according to our prior studies [15,16,17]. The presence of FS can be used as an independent predictor of an unfavorable prognosis in patients with CAD, both in the perioperative period and early survival period after CABG.

Despite the fact that the guidelines claim that the presence of FS should be assessed routinely in elderly and senile patients, there are no clear recommendations regarding the preoperative period of CABG [18]. Presumably, this is due to difficulties associated with diagnosing this syndrome, and the lack of protocol for identifying and managing high-risk frail patients [19]. The authors of the current study have not managed to find a distinguished approach to diagnosing FS. Unfortunately, this kind of research topic remains understudied. The strength of relationships obtained with tools for the identification of FS and unfavorable outcomes after surgery differs according to a number of studies and depends on the time frame in which the diagnosis of frailty is established. Thus, various approaches for the detection of FS indicate an average effect size of frailty at the baseline upon the incidence of adverse outcomes (d = 0.4–0.6), whereas the effect size is larger in the early period (d = 0.1–0.8) [20]. The study carried out by D.I. McIsaac et al. [12], which included elderly patients undergoing scheduled non-cardiac surgery, analyzed the role of FS in the development of complications and adverse outcomes using the L.P. Fried frailty phenotype assessment and the Clinical Frailty Scale. The results revealed no differences in the sensitivity or specificity between these approaches. A similar study by Z. Cooper et al. [21] also found no differences between the frailty phenotype and the frailty scaling when predicting the length of hospital stay, development of complications, and adverse outcomes.

In our study, we used the “PRISMA-7” questionnaire as a tool for screening FS, and it yielded different results in terms of sensitivity and specificity [22,23]. Nevertheless, based on the results obtained during FS screening with “PRISMA-7”, the present study revealed significant differences in the incidence of complications and adverse outcomes in the perioperative period and early survival period after CABG in frail patients. This can serve as a justification for conducting additional studies to develop an algorithm for the diagnosis and management of frail patients in the perioperative period of CABG in order to improve the course of the early postoperative period and reduce the risk of mortality within a year after surgery [24].

One of the first studies that analyzed the effect of FS on the prognosis of cardiac surgery patients was a cohort study by D.H. Lee et al. [25], which included 3826 patients. The study showed that frail patients (4.1%) had a higher risk of mortality both in the in-hospital period (OR 1.8 and 95% CI 1.1–3.0) and 2 years after surgery (OR 1.5 and 95% CI 1.1–2.2). A study carried out by K. Clark et al. [26] revealed a significant relationship between FS and high mortality rates after open heart surgery. In a study by Y. Imaoka et al. [27], the analysis of the data obtained showed that FS was significantly associated with an unfavorable prognosis (*p* = 0.004) and with a high mortality rate of patients undergoing surgery (*p* < 0.001). In a study by T.Z. Ali et al. [28], it was found that frailty can be used for identifying patients at higher risk of postoperative complications. Several systematic reviews performed earlier have also shown that FS, detected by various diagnostic methods, is an independent predictor of adverse outcomes of surgical intervention [10,11]. Thus, the results of our study confirm that frailty at baseline contributes significantly to the risk of adverse outcomes and complications, both in the perioperative period and early survival period after CABG. However, there is still no single “gold standard” method for diagnosing FS in the preoperative period.

Specialists all over the world have favorably received and used the European System for Cardiac Operative Risk Evaluation (EuroSCORE) since its first introduction in 1999 [29]. Using a large and tightly controlled patient database drawn from across Europe, the system uses logistic regression to identify and give appropriate weight to various risk factors related to mortality in adult cardiac surgery. American and European patient populations were compared in terms of demographic characteristics, incidence of surgical procedures performed and prevalence of risk factors. The simple, additive EuroSCORE model was then tested on two groups of patients from the STS database: all patients who underwent adult cardiac surgery in 1995 and in the period from 1998 to 1999. The first group of patients was chosen because EuroSCORE was developed using the 1995 European patient cohort and the second group was chosen because of greater similarity between the American and European datasets, in terms of recency and relevance. The datasets included various factors that increase surgical risk, accounting for age of patients and comorbidities; however, other important predictors such as physical, cognitive, and social factors that underlie the biological aging were not addressed. Although the recommendations of different medical societies mention the need to assess the presence of frailty in routine clinical practice in elderly and very elderly patients, there are no universal recommendations regarding the preoperative period of CABG. This is probably due to the difficulties of diagnosing this syndrome and the lack of an algorithm for the detection and management of “fragile” high-risk patients.

Several clinical studies assessing the impact of FS on early prognosis of patients undergoing cardiac surgery indicate that the preoperative assessment of frailty increases the sensitivity of other models of assessing cardiac risk in relation to the prediction of one-year mortality. In a study by S.H. Sündermann et al. [30], an additional assessment of FS in the preoperative period, coupled with an assessment of cardiac surgical risk with the STS and EuroSCORE scale, significantly increased the sensitivity of predicting one-year mortality after surgery. The authors of this study also suggest that frailty is an indicator of biological aging of the human body, since there were no correlations between FS and the age ranges in the sample. In another study, frailty was an independent predictor of medium-term mortality after cardiac surgery (relative risk (RR) 2.05, 95% CI 1.43–2.85; RR 3.05, 95% CI 1.83–5.06), and increased the sensitivity of the EuroSCORE II prognostic scale (*p* = 0.028) [31].

Given the exponential growth of the elderly and senile population, the problem of frailty becomes more relevant every year. First of all, FS lowers the quality of life of patients, and it is a predictor of unfavorable hospital outcomes and 1-year mortality. Thus, in several clinical studies, the primary role of frailty in the development of various complications and adverse outcomes after surgery has been proven. However, there are no data on what can be used as the most accurate method for diagnosing FS in the preoperative period. Taking into account the increasing incidence of frailty, further clinical studies are necessary to develop diagnostic and preventive measures in order to reduce early postoperative mortality.

To choose an appropriate method of myocardial revascularization, it is preferable to use the “Heart-Team” approach. This is primarily due to the high morbidity and mortality associated with CAD. The chosen treatment modality should be effective and safe for patients with stable CAD, and must take into account a wide range of patient ages, the full range of comorbidities, surgical risks, and patient preferences. At the same time, the chosen type of myocardial revascularization should prolong the patient’s life and improve its quality. The main approaches to choosing a method of myocardial revascularization are reflected in the latest recommendations and are based on taking into account the severity of coronary lesions and the presence of comorbid pathology (HF, rhythm disturbances, and DM) [32]. Specific factors such as frailty, which leads to increased vulnerability to stress and an increased risk of morbidity and mortality after open cardiac surgery, should also be taken into account during decision making [33]. These factors can significantly affect the “Heart-Team” decisions regarding the treatment and management of patients with FS and CAD.

## 5. Conclusions

The presence of frailty can be used as an independent predictor of an unfavorable prognosis in patients with CAD, both in the perioperative and early survival period after CABG. It should be taken into account during surgical risk assessment.

## Figures and Tables

**Figure 1 diagnostics-14-01419-f001:**
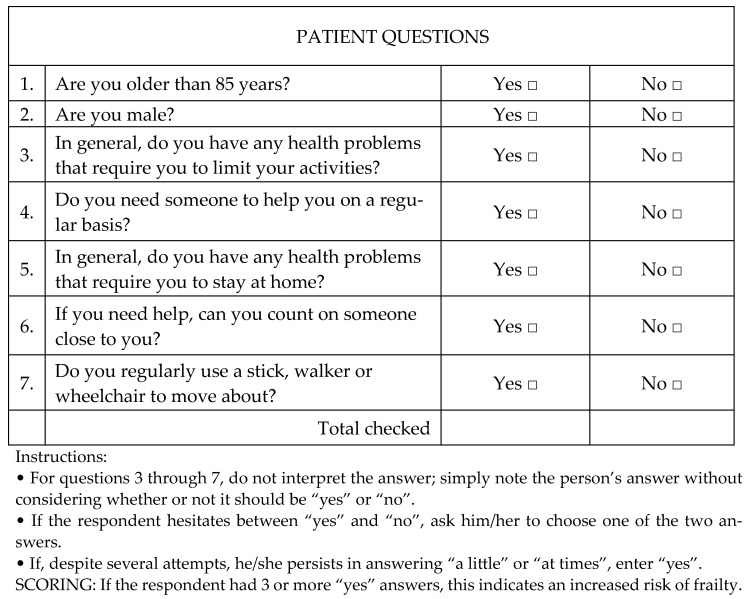
“PRISMA-7” questionnaire.

**Figure 2 diagnostics-14-01419-f002:**
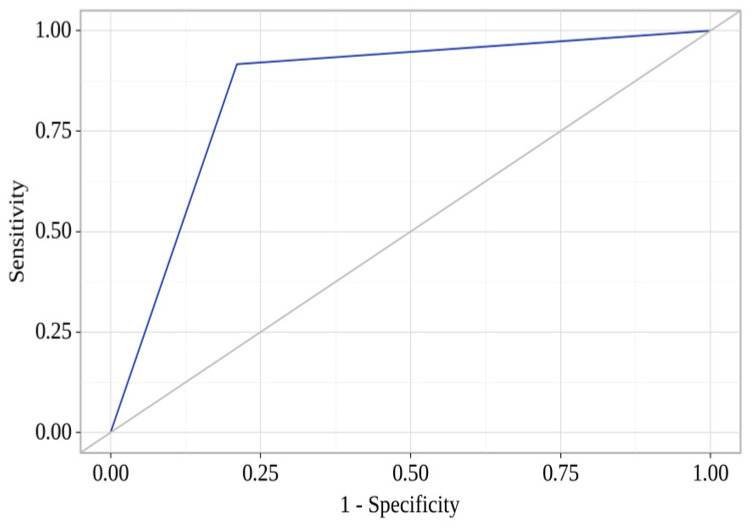
ROC curve for assessing the risk of mortality within the first year after coronary artery bypass grafting from the value of the logistic function p.

**Table 1 diagnostics-14-01419-t001:** Baseline clinical and anamnestic characteristics of patients with stable coronary artery disease.

Parameter	Patient Characteristics *(n =* 387)
Mean age, years (Me [Q_1_; Q_3_])	65.0 [59.0; 69.0]
Men, *n* (%)	283 (73.1)
BMI, kg/m^2^ (Me [Q_1_; Q_3_])	29.3 [26.6; 32.1]
Smoking, *n* (%)	178 (45.9)
Duration of CAD, years (Me [Q_1_; Q_3_])	2.0 [1.0; 5.0]
FC III-IV angina, *n* (%)	70 (18.1)
PICS, *n* (%)	221 (57.1)
Atherosclerosis of the extracranial brachiocephalic arteries, *n* (%)	225 (58.1)
FC III-IV HF, *n* (%)	32 (8.3)
PCI in history, *n* (%)	73 (18.9)
TIA/Stroke in history, *n* (%)	43 (11.1)
CEE in history, *n* (%)	4 (1.0)
Ulcers, *n* (%)	13 (3.4)
AH, *n* (%)	323 (83.5)
AFL/AFib in preoperative period, *n* (%)	44 (11.4)
Pacemaker implantation in history, *n* (%)	2 (0.5)
DM II type, *n* (%)	98 (25.3)
Impaired glucose tolerance, *n* (%)	74 (19.1)
PAD, *n* (%)	124 (32.0)
CRF, *n* (%)	43 (11.1)
Bone fractures in history, *n* (%)	30 (7.8)
Femoral neck fractures in family history, *n* (%)	13 (3.4)
Rheumatoid arthritis, *n* (%)	2 (0.5)
Secondary osteoporosis, *n* (%)	7 (1.8)
Left main coronary artery stenosis, *n* (%) Number of diseased coronary arteries, *pcs* (Me [Q1; Q3])	86 (22.2) 2.0 [2.0; 3.0]

Note: BMI—body mass index; PCI—percutaneous coronary intervention; TIA—transient ischemic attack; CEE—carotid endarterectomy; AFib—atrial fibrillation; AFL—atrial flutter; CRF—chronic renal failure; and PICS—postinfarction cardiac sclerosis.

**Table 2 diagnostics-14-01419-t002:** Clinical and anamnestic characteristics of frail patients with stable coronary artery disease.

Parameter	Group of Patients without Frailty, n_0_ = 300 (77.5%)	Group of Frail Patients, n_1_ = 87 (22.5%)	*p* (Cramer’s V)
Mean age, years (Me [Q1; Q3])	65.0 [59.0; 70.0]	64.0 [59.5; 67.0]	0.128
Men, *n* (%)	209 (69.9)	74 (84.1)	0.008 * (0.134)
BMI, kg/m^2^ (Me [Q_1_; Q_3_])	29.1 [26.6; 31.8]	30.1 [26.1; 32.7]	0.459
Smoking, *n* (%)	130 (43.5)	48 (54.5)	0.067
Duration of CAD, years (Me [Q_1_; Q_3_])	2.0 [1.0; 5.0]	4.0 [1.0; 5.5]	0.092
Duration of AH, years (Me [Q_1_; Q_3_])	10.0 [5.0; 15.0]	7.0 [3.0; 16.5]	0.459
FC III-IV Angina, *n* (%)	55 (21.1)	11 (15.7)	0.319
PICS, *n* (%)	162 (54.2)	59 (67.0)	0.032 * (0.109)
Atherosclerosis of the extracranial brachiocephalic arteries, *n* (%)	188 (62.9)	44 (50.0)	0.030 * (0.110)
III-IV HF, *n* (%)	22 (8.3)	6 (7.5)	1.000 ^1^
PCI in history, *n* (%)	55 (18.3)	22 (25.3)	0.153
Stroke in history, *n* (%)	30 (10.0)	6 (6.8)	0.412 ^1^
TIA in history, *n* (%)	5 (1.7)	2 (2.3)	0.660 ^1^
CEE in history, *n* (%)	3 (1.0)	1 (1.1)	1.000 ^1^
Ulcers, *n* (%)	9 (3.0)	9 (10.2)	0.009 ^1^* (0.144)
AH, *n* (%)	247 (82.3)	79 (90.8)	0.056
AFib in preoperative period, *n* (%)	32 (12.7)	8 (11.6)	1.000 ^1^
AFL in preoperative period, *n* (%)	4 (1.7)	2 (2.6)	0.636 ^1^
Pacemaker implantation in history, *n* (%)	1 (0.3)	1 (1.1)	0.404 ^1^
Type II DM, *n* (%)	82 (27.4)	18 (20.5)	0.189
PAD, *n* (%)	87 (29.1)	37 (42.0)	0.022 * (0.116)
CRF, *n* (%)	41 (13.7)	7 (8.0)	0.161
Apnea, *n* (%)	10 (3.3)	1 (1.1)	0.468 ^1^
Bone fractures in history, *n* (%)	17 (5.7)	13 (14.9)	0.010 ^1^* (0.145)
Femoral neck fractures in family history, *n* (%)	11 (3.7)	2 (2.3)	0.741 ^1^
Rheumatoid arthritis, *n* (%)	1 (0.3)	1 (1.1)	0.400 ^1^
Secondary osteoporosis, *n* (%)	8 (2.7)	2 (2.3)	1.000 ^1^
EuroSCORE II (Me [Q_1_; Q_3_])	1.6 [1.2; 2.0]	1.4 [1.1; 1.7]	0.063

Note: ^1^ Fisher’s exact test and * statistically significant differences (*p* ≤ 0.050). BMI—body mass index; CAD—coronary artery disease; AH—arterial hypertension; FC—functionality class; PICS—postinfarction cardiac sclerosis; HF—heart failure; PCI—percutaneous coronary intervention; TIA—transient ischemic attack; CEE—carotid endarterectomy; AFib—atrial fibrillation; AFL—atrial flutter; DM—diabetes mellitus; PAD—peripheral artery disease; and CRF—chronic renal failure.

**Table 3 diagnostics-14-01419-t003:** Intraoperative characteristics of patients undergoing open myocardial revascularization.

Parameter	Group of Patients without Frailty, n_0_ = 300 (77.5%)	Group of Frail Patients, n_1_ = 87 (22.5%)	*p*
CPB time, min (Me [Q_1_; Q_3_])	76.0 [65.0; 97.0]	77.0 [64.0; 86.5]	0.920
Aortic cross-clamping, min (Me [Q_1_; Q_3_])	52.0 [40.0; 64.0]	51.0 [40.0; 58.0]	0.432
Number of grafts, pcs (Me [Q_1_; Q_3_])	2.0 [2.0; 3.0]	2.0 [2.0; 3.0]	0.389
Minimal systolic BP, mmHg (Me [Q_1_; Q_3_])	100.0 [92.0; 107.0]	98.0 [96.0; 104.0]	0.327
MV time, min (Me [Q_1_; Q_3_])	645.0 [548.0; 825.0]	640.0 [550.5; 879.0]	0.787
Blood loss on the first day after surgery, mL (Me [Q_1_; Q_3_])	300.0 [250.0; 400.0]	350.0 [300.0; 500.0]	0.092
ICU length of stay, hours (Me [Q_1_; Q_3_])	21.5 [19.5; 24.5]	23.0 [19.0; 43.0]	0.310

Note: BP—blood pressure; MV—mechanical ventilation; and ICU—intensive care unit.

**Table 4 diagnostics-14-01419-t004:** Outcome and complications of surgery depending on the presence of frailty syndrome.

Outcome and Complications	Group of Patients without Frailty, n_0_ = 300 (77.5%)	Group of Frail Patients, n_1_ = 87 (22.5%)	*p*
Various complications, *n* (%)	57 (19.6)	30 (35.7)	0.002 * (0.159)
Death, *n* (%)	1 (0.3)	4 (4.5)	0.010 ^1^* (0.156)
Myocardial infarction, *n* (%)	2 (0.7)	2 (2.3)	0.218 ^1^
Stroke, *n* (%)	4 (1.3)	5 (5.7)	0.031 ^1^* (0.122)
Infectious complications, *n* (%)	11 (3.7)	6 (6.8)	0.238 ^1^
HF with extended inotropic support, *n* (%)	20 (6.7)	9 (10.2)	0.259 ^1^
Rhythm disturbances, *n* (%)	31 (10.5)	17 (19.5)	0.025 * (0.115)
Pleural puncture, *n* (%)	5 (1.7)	–	0.593 ^1^
Sternal diastasis, *n* (%)	1 (0.3)	1 (1.1)	0.404 ^1^
MODS, *n* (%)	1 (0.3)	2 (2.3)	0.131 ^1^
Death within a year after CABG, *n* (%)	2 (0.6)	10 (11.5)	˂0.001 ^1^* (0.290)

Note: ^1^ Fisher’s exact test and * Statistically significant differences (*p* ≤ 0.050). HF—heart failure; MODS—multiple organ dysfunction syndrome.

## Data Availability

Data is unavailable due to privacy and ethical restrictions.

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
