# Peer review of "Frailty as an Independent Predictor of Adverse Outcomes in Patients Undergoing Direct Myocardial Revascularization"

_diagnostics, 2024, doi:10.3390/diagnostics14131419_

Round 1

Reviewer 1 Report

Comments and Suggestions for Authors

Many thanks for the opportunity to review this manuscript. The paper includes 387 patients who had cabg and were assessed with PRISMA-7 questionnaire for frailty and its impact on adverse outcomes

The paper is well written with appropriate references and literature search. Appropriate statistical methodology has been employed and results are well supported and illustrated in tables.

Line 10-11 ….outcomes in the perioperative and long-term period of coronary artery bypass grafting (CABG).

As there was only 1 year follow up, please rephrase this as ….perioperative period and early survival after cabg. Please replace ‘long term period’ throughout the manuscript.

Line 27, 28 – please rephrase the conclusion.

Please explain all abbreviations used in Table 2

Please explain PRISMA-& in methods and add a table /supplementary table of the questionnaire

Please define frailty and frailty syndrome in introduction

Mention ROC analysis in methods – there is no mention.

Mention ROC curve analysis in abstract results

Discussion

Please elaborate on the STS and EuroSCORE – how were they devised and why they do not assess frailty in cardiac surgery patients.

Please comment on utility of using these scoring systems for elderly higher risk patients with multiple co-morbidities not captured by the scores (eg liver disease, cirrhosis, anemia, low body weight etc)

Conclusions - Line 270-279. Please rewrite and shorten the conclusions. Conclusions should be based only on the findings of the study.

Author Response

Dear Reviewer,

Thank you for carefully reading our article and for your valuable comments. All comments were taken into account, the necessary changes were made to the text of the article (yellow).

Best regards,

Kristina

Reviewer 2 Report

Comments and Suggestions for Authors

This study evaluates the outcomes after CABG in patients with and without frailty. They found higher neurologic event in early postoperative period and higher mortality after one-year follow-up in the frail group.

After careful reading of the manuscript, some following considerations can be made:

1.     Abstract: I think statistical method is not to be described in abstract section. (Line 15-16 “Statistical processing of the results…”). Also you need to change comma “,” into point “.” if you write p<0.05.(Line 177)

2.     Abstract: some English phrases are not correct: “long-term period of coronary artery bypass grafting”->for example ”long-term outcome after coronary artery bypass grafting”. Further, I should say, 1-year follow-up is not “long-term follow-up”. It might be changed. 

“Thus” might be changed. Also you wrote “intraoperative difference was revealed” and which parameter?

3.     Abstract: for postoperative rhythm disturbances and TIA/stroke rate, you wrote frail patients vs non-frail patients. But 1-year mortality was inverted, I guess (non- frail vs frail patients; 0.6% vs 11.5%?). Further, you could add the early death because there was significant difference. 

4.     Abstract: please add how many patients in frail group and in non-frail group.

5.     Results: how did you detect cerebral atherosclerosis? Using CT or MRI before surgery? If you exclude any preoperative central nervous disease, please add the information.

6.     Methods: please add the criteria of PRISMA-7 as a figure. It might be grate for readers to know which parameters were used.

7.     Methods: please add the description about ROC-curve analysis.

8.     Results Line 127: there is a contradiction. P=0.008 is not statistically significant.

9.     Table 2: please add the description of abbreviations.

10.  Results Line 130: Do you mean “acute” myocardial infarction or “previous”?

11.  Results/Table: please add the number of diseased coronary arteries, if you mention the number of grafts in the intraoperative parameter. And also please add how many bypass were done in intraoperative section. 

12.  Results Line 154: what is the definition of rhythm disturbance? Also please add the definition of TIA and stroke in methods section.

13.  Discussion: using this result, what do you suggest clinically? For frail patients PCI might be better? 

14.  Conclusion: it is just assumption of the former studies. Please write your conclusion using your result.

15.  Abbreviation: you wrote AF both for atrial flatter and fibrillation. It should be changed. Further, some other abbreviations are lacking, including “AH” etc.

Comments on the Quality of English Language

Abstract: some English phrases are not correct: “long-term period of coronary artery bypass grafting”->for example ”long-term outcome after coronary artery bypass grafting”. Further, I should say, 1-year follow-up is not “long-term follow-up”. It might be changed. 

“Thus” might be changed. Also you wrote “intraoperative difference was revealed” and which parameter?

Author Response

(The authors gave the same response as above.)

Round 2

Reviewer 2 Report

Comments and Suggestions for Authors

I appreciate your corrections. Some following considerations can be still made.

Comment 1: Line 87-93: I appreciate that you added the criteria of the questionnaire “PRISMA-7”. However, in my opinion, it would be nice as a figure format. Please reconsider it. Further, I wonder this criterion of the questionnaire. Then, if a male patient older than 85 years has a symptom of ischemic heart disease, the case might be categorized into a frail group. Please mention in limitation this issue. It might lead a bias on your study. 

Comment 2: Line 104-113: this text seems like “copy-and-paste”. It is not acceptable! And is not a definition of criteria of postoperative heart rhythm disorder. 

Comment 3: Line 139-140: how do you define “atherosclerosis of the main artery of the head? Did you get all CT-scan preoperatively?

Comment 4: you added a comment in discussion as a consequence of your study. However, the result showed not only mortality of 1-year follow-up but also early postoperative morbidity. It could not be improved just only by postoperative rehabilitation programs. Please mention also a preoperative assessment of operability and decision between “Heart-Team”.

Author Response

Sorry for our carelessness! We tried to make corrections (green)!
